# Domain-Conditioned Transformer for Fully Test-time Adaptation

## ABSTRACT

Fully test-time adaptation aims to adapt a network model online based on sequential analysis of input samples during the inference stage. We observe that, when applying a transformer network model into a new domain, the self-attention profiles of image samples in the target domain deviate significantly from those in the source domain, which results in large performance degradation during domain changes. To address this important issue, we propose a new structure for the self-attention modules in the transformer. Specifically, we incorporate three domain-conditioning vectors, called domain conditioners, into the query, key, and value components of the self-attention module. We learn a network to generate these three domain conditioners from the class token at each transformer network layer. We find that, during fully online test-time adaptation, these domain conditioners at each transform network layer are able to gradually remove the impact of domain shift and largely recover the original self-attention profile. Our extensive experimental results demonstrate that the proposed domain-conditioned transformer significantly improves the online fully test-time domain adaptation performance and outperforms existing state-of-the-art methods by large margins.

## CCS CONCEPTS

• **Computing methodologies** → **Transfer learning**; **Online learning settings**; **Computer vision**.

## KEYWORDS

Test-time Adaptation, Domain-Conditioned Transformer

## 1 INTRODUCTION

Transformers have achieved remarkable success in various machine learning tasks. However, their performance often degrades significantly when being tested in new domains due to the data distribution shifts [40] between the training data in the source domain and the test data in the target domain [36]. Source-free unsupervised domain adaptation (UDA) [26, 30, 48, 52] aims to adapt network models without access to source-domain samples. Nevertheless, these approaches require complete access to the entire target dataset and retraining of the source model for multiple epochs, making them impractical for real-world applications. Recently developed test-time adaptation (TTA) methods exhibit promising capabilities in adapting pre-trained models to unlabeled data during testing [29, 37, 46, 49, 51]. There are two major types

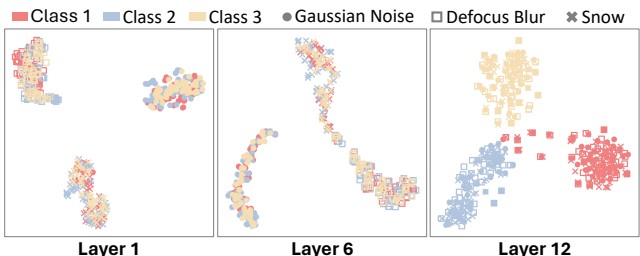

**Figure 1: Visualization of output class tokens across various layers of our adapted ViT-B/16 network in ImageNet-C dataset. In layer 1, the features exhibit domain-separability and class-inseparability due to the presence of domain shift, with a considerable distance between domains and a small distance between classes. Our DCT method effectively mitigates the influence of domain shift over successive layers. Consequently, the domain distance decreases while the class distance increases, leading to the features domain-inseparable yet class-separable across the layers of the Domain-Conditioned Transformer.**

of TTA methods: (1) test-time training (TTT) [10, 35, 46] and (2) fully test-time adaptation [37, 38, 49, 51], depending on whether source domain data is accessed or not. In this study, we focus on fully test-time adaptation.

For fully test-time adaptation, the TENT method [51] updates the batch normalization module by minimizing entropy loss. The MEMO method [58] optimizes the entropy of averaged predictions over multiple random augmentations of input samples. Meanwhile, the VMP method [22] introduces perturbations into model parameters based on variational Bayesian inference. Note that these approaches assume a sufficiently large number of samples in a mini-batch and a well-balanced label distribution in each mini-batch of the target domain. However, this assumption does not always hold in practice. To address this challenge, the SAR method [38] proposes a sharpness-aware and reliable optimization scheme, which eliminates samples with significant gradients and encourages model weights to converge to a flat minimum. The TTN method [31] optimizes interpolation weights during the post-training phase, requiring access to labeled source data. The RoTTA method [56] introduces robust batch normalization through category-balanced sampling.

Recently, transformer-based methods have achieved remarkable success in various machine learning tasks due to their powerful self-attention capabilities. In this work, we propose to explore how transformer networks can be successfully adapted to new domains during the testing stage. During our experiments, we find that when transformer models are applied to new domains, their self-attention distance profiles, defined as the spatial distribution of self-attention between tokens, for image samples in the target domain deviate significantly from those in the source domain. Note that the self-attention is one of the core modules in the transformer network

design. Once this self-attention profile has been perturbed by the domain changes or corruptions, the performance of the transformer model will degrade significantly. A research question arises: *how do we remove these perturbations caused by the domain shifts from the self-attention profile so as to improve the test-time adaptation performance of transformer models?*

To address this challenge, we propose to explore a new approach, called *domain-conditioned transformer*, for fully test-time adaptation of transformer models. Specifically, we introduce three domain-conditioning vectors, called *domain conditioners*, into the query, key, and value components of the self-attention modules. These domain conditioners are designed to capture domain-specific perturbation information and remove these perturbations layer by layer. We learn a domain conditioner generation network to generate these domain conditioners from class tokens of the previous network layers, containing both semantic and domain information. We observe that, our proposed approach gradually mitigates the impact of domain shift. As shown in Figure 1, the domain information is gradually removed and the class information is enhanced across the domain-conditioned transformer layers. This gradual adjustment process facilitates the recovery of the original self-attention profile, allowing the model to maintain its performance across diverse domains. Our extensive experimental results demonstrate that the proposed domain-conditioned transformer significantly improves the online test-time domain adaptation performance and outperforms existing state-of-the-art methods by large margins.

## 2 RELATED WORK

This work is related to test-time adaptation, source-free unsupervised domain adaptation, parameter efficient transfer learning, and prompt learning.

### 2.1 Test-time Adaptation

Test-time adaptation (TTA) aims to adapt a pre-trained source model to unlabeled data with domain shift during inference. There are two major approaches: *test-time training* [10, 35, 46] and *fully test-time adaptation* [37, 38, 51]. [46] proposes the first test-time training (TTT) method where feature extractor network parameters are updated using a self-supervised loss on a proxy learning task. The TTT++ method [35] improves this approach with a feature alignment strategy based on online moment matching. Extending this line of research, the TTT-MAE method [10] incorporates a transformer backbone and replaces self-supervision with masked auto-encoders [15]. Note that all these TTT methods require specialized training in the source domain.

In contrast, fully test-time adaptation methods fine-tune pre-trained models during inference without access to the source data. The TENT method [51] proposes fully test-time adaptation by fine-tuning Batch Normalization (BN) layers. The NHL method [49] learns early-layer representations in an unsupervised manner, drawing inspiration from neurobiology-inspired Hebbian learning. Methods have also been developed to update the model inputs instead of the network parameters. For example, the DDA method [11] projects the input data from the target domain into the source domain based on a diffusion model during testing. The method proposed by [9] modifies the target inputs by learning image-level

visual prompts, keeping source model parameters frozen during testing. It has been noted that existing online model updating methods suffer from performance degradations due to sample imbalances and distribution shifts. To address this issue, SAR [38] proposes to eliminate noisy samples with high gradients and perform flattening of model weights towards a minimum, thereby enhancing the robustness of the model. DELTA [59] uses moving averaged statistics to perform the online adaptation of the normalized features.

### 2.2 Source-free Unsupervised Domain Adaptation

Source-free unsupervised domain adaptation (source-free UDA) aims to adapt the model trained on the source domain to the unlabeled target domains without leveraging the source data [19, 24, 26, 27, 30, 45, 55]. Pseudo-labeling [24] methods assign a class label for each unlabeled target sample and uses the label for the supervised learning objective. The SHOT method [30] computed pseudo labels through the nearest centroid classifier and optimized the model with information maximization criteria. The KUDA method [45] utilized the prior knowledge about label distribution to refine model-generated pseudo labels. The SFDA-DE method [5] aligned domains by estimating source class-conditioned feature distribution. The HCL method [19] proposed a solution for addressing the lack of source data by introducing both instance-level and category-level historical contrastive learning. The DIPE method [52] focuses on exploring the domain-invariant parameters of the model, rather than trying to learn domain-invariant representations. These source-free methods are offline, requiring access to the complete test dataset. It also costs a number of epochs for model adaptation. In contrast, our fully online test time adaptation adapts the given source model on the fly during testing which only accesses the test samples once.

### 2.3 Parameter Efficient Transfer Learning

As the model size grows rapidly with the development of foundation models, there has been a growing interest among researchers focusing on Parameter Efficient Transfer Learning [18, 25]. This area of study focuses on adapting large-scale pre-trained models to different downstream tasks with minimal modification of parameters. These methods strategically select a subset of pre-trained parameters and, if needed, introduce a limited number of additional parameters into a pre-trained network. These selected parameters are updated specifically for new tasks, while the majority of the original model parameters are frozen to ensure efficiency and effectiveness. For instance, the method proposed by [32] introduces learnable vectors to rescale keys, values in attention mechanisms, and inner activations in position-wise feed-forward networks through element-wise multiplication. *Diff pruning* [14] learns an adaptively pruned task-specific "diff" vector extending the original pre-trained parameters. BitFit [57] employs sparse fine-tuning, where only the bias terms (or a subset of them) are modified. AdaptFormer [4] adapts pre-trained Vision Transformers (ViTs) for various vision tasks by replacing the original MLP block with a trainable down-up bottleneck module. In contrast, LST [47] introduces a separate ladder side network, a smaller network that uses intermediate activations as inputs based on shortcut connections, rather than inserting additional parameters inside the backbone networks. It

should be noted that all of these parameter efficient transfer learning methods adapt pre-trained models to downstream tasks through supervised learning.

## 2.4 Prompt Learning

The notion of prompts originated in Natural Language Processing (NLP), where linguistic instructions, known as prompts, are added to input text to guide pre-trained language models in specific downstream tasks [33]. Recent methods [25, 28, 34] represent prompts as task-specific vector inputs and optimize them directly through error back-propagation. These prompt-tuning methods learn prompts from downstream data within the input embedding space, requiring fewer parameters to be updated during the adaptation process.

Prompt learning has been successfully applied to vision-language models [13, 43, 60, 61]. CoOp [61] fine-tunes the prompt of the text encoder in CLIP [41]. CoCoOp [60] conditions the learned prompt on the model's input data to address out-of-distribution issues. TPT [43] optimizes the prompt of CLIP's text encoder during test time, enhancing generalization performance by minimizing entropy with confidence selection. DAPL [13] constructs a prompt comprising domain-agnostic context, domain-specific context, and class label for the text encoder using a contrastive objective to disentangle semantic and domain representations. It's crucial to note that these prompt methods are based on text encoders in Vision-Language Models, whereas our approach exclusively focuses on the Vision Transformer (ViT) encoder.

In addition to text prompt learning, techniques have emerged for learning visual prompts in computer vision tasks. The method proposed by [9] focuses on continuous TTA tasks, where domain-specific and domain-agnostic prompts are learned and attached to target input images on a per-pixel basis. Meanwhile, BlackVIP [39] learns individual prompts for each image through a neural network, without requiring prior knowledge about pre-trained model architectures and parameters. Visual Prompt Tuning (VPT) [12, 21] introduces task-specific learnable parameters into the input sequence of each ViT encoder layer while keeping the pre-trained transformer encoder backbone frozen during downstream training. This approach has found applications in transfer learning for image synthesis [44]. Note that the VPT method introduces a large number of new tokens. It substantially increases the computational complexity of the self-attention mechanism, as the computational complexity of self-attention is quadratic to input token size.

## 2.5 Unique Contributions

In this work, we propose to explore fully online test-time adaptation of transformer models by designing adaptable self-attention modules. Compared to existing work, the major contributions of this work can be summarized as follows: (1) We introduce a new design of the self-attention module in the transformer networks, which is able to capture the domain-specific characteristics of test samples in the target domain and is able to clean up the domain shift perturbations in the test samples. (2) We learn a lightweight neural network called domain-conditioner generator to generate the domain conditioners from the class token at each layer, enabling the transformer model to better align its self-attention profiles with the source domain. (3) Our experimental results demonstrate that our

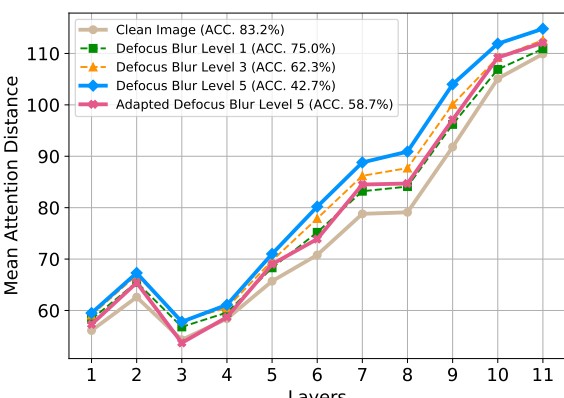

**Figure 2: Size of the attended area by transformer network depth. Each dot on the figure represents the mean attention distance calculated across 128 example images, considering all heads at a specific layer.**

proposed domain-conditioned transformer is able to significantly improve the online domain adaptation performance of transformer models, outperforming the state-of-the-art method in fully test-time domain adaptation across multiple popular benchmark datasets and test conditions.

## 3 METHOD

In this section, we present our method of Domain-Conditioned Transformer (DCT) for fully test-time adaptation.

## 3.1 Method Overview

Suppose we have a model $\mathcal{M} = f_{\theta_s}(y|X_s)$ with parameters $\theta_s$, successfully trained on source data $\{X_s\}$ with corresponding labels $\{Y_s\}$. During fully test-time adaptation, we are provided with target data $\{X_t\}$ along with unknown labels $\{Y_t\}$. Our objective is to adapt the trained model online in an unsupervised manner during testing. In this scenario, we receive a sequence of input sample batches $\{\mathbf{B}_1, \mathbf{B}_2, ..., \mathbf{B}_T\}$ from the target data $\{X_t\}$. It should be noted that, during each adaptation step $t$, the network model can only rely on the $t$-th batch of test samples, denoted as $\mathbf{B}_t$. Following the *wild* test-time adaptation setting outlined in SAR [38], it's possible that each mini-batch $\mathbf{B}_t$ may contain only one sample, and the samples within the mini-batch can be imbalanced.

When adapting transformer-based models to new domains, we observe that their self-attention distance profiles, defined as the spatial distribution of self-attention between tokens, for image samples in the target domain deviate significantly from those in the source domain. Let $\{\Omega_i^l | 1 \le i \le N\}$ be the set of $N$ embedding tokens at the network layer $l$, and $\mathbb{C} = [c_{ij}^l]_{N \times N}$ be their self-attention weight matrix. Let $\mathbb{D} = [d_{ij}^l]_{N \times N}$ be the distance matrix for these tokens where $d_{ij}^l$ represents the pixel distance in the original input image between tokens $i$ and $j$. The attention distance [6, 42] for network layer $l$ is then defined as

$$d(l) = \sum_{i,j} c_{ij}^l \cdot d_{ij}^l. \tag{1}$$

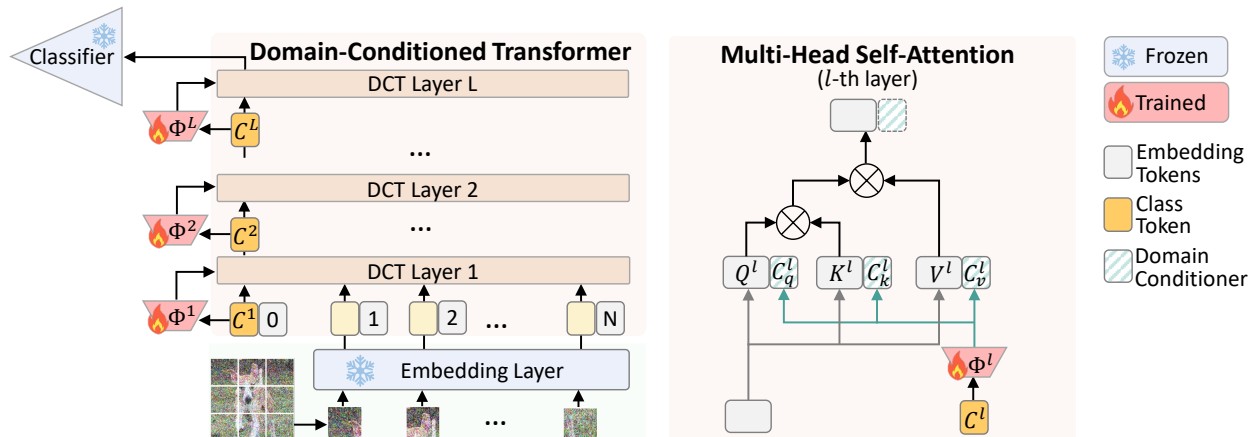

**Figure 3: An overview of the proposed DCT method. During inference in the target domain, the domain conditioners generator $\Phi^l$ and LN layers are updated before making a prediction given each mini-batch testing sample. The domain-conditioned transformer (Left). The details of the self-attention head in each layer (Right).**

The attention distance $d^l$ for all network layers is referred to as the self-attention profile, denoted as $\{d(l)\}$. In Figure 2, we plot the attention profile for the clean image, images with different levels of domain corruptions (Defocus Blur), and the attention profile recovered by our DCT method. Conceptually, this attention distance is similar to the receptive field size in Convolutional Neural Networks (CNNs). It indicates that lower layers of the ViT model tend to focus on local regions more, as evidenced by a lower mean attention distance. In contrast, higher layers primarily integrate global information, leading to a higher attention distance. When there is a data distribution shift during testing in the target domain, the attention distance distribution is shifted. As illustrated in Figure 2, as the level of image corruption increases, the corresponding attention distance becomes large. Once this self-attention profile has been perturbed by the domain changes or corruptions, the performance of the transformer model will degrade significantly. It can be seen that, using the DCT method, the attention distance profiles of the corrupted images can be largely recovered to their minimum level (Level 1), approaching the attention distance profile of the clean image. Meanwhile, the Attention Rollout for the target samples exhibits improved focus compared to the source model, as illustrated in Figure 6.

In this work, we propose to introduce a new self-attention structure that is able to capture the domain perturbations and gradually remove them from the image features. As shown in Figure 3, at layer $l$ of the proposed domain-conditioned transformer (DCT), we append three domain-conditioning vectors, $[C_q^l, C_k^l, C_v^l]$, into the query, key, and value components $[Q^l, K^l, V^l]$ of its self-attention module. At each transformer network layer, we learn a lightweight neural network (domain-conditioner generator $\Phi^l$) to generate these three domain conditioners $[C_q^l, C_k^l, C_v^l]$ from the class token $C^l$. During fully test-time adaptation, $\Phi^l$ is updated during the inference process. These domain conditioners at each transform network layer are able to gradually remove the impact of domain shift and significantly recover the original self-attention profile. In the following section, we explain the proposed DCT method in more detail.

## 3.2 Domain-Conditioned Self-Attention

Self-attention mechanisms have demonstrated remarkable performance in various computer vision tasks by capturing correlation between image patches. The output matrix of self-attention is defined as:

$$\text{Attention}(Q, K, V) = \text{softmax}\left(\frac{QK^\top}{\sqrt{d}}\right)V, \quad (2)$$

where $d$ represents the dimensions of the query, key, and value. For convenience, we consider layer $l$ of the network and omit the superscript $l$ here in this section. The self-attention weights are computed from the correlation between patch embeddings. Certainly, the self-attention distance profile defined in the previous section changes when the input image is perturbed by domain shifts. Write the query $Q \in \mathbb{R}^{n \times d}$, key $K \in \mathbb{R}^{n \times d}$, and value $V \in \mathbb{R}^{n \times d}$ matrices of the self-attention mechanism for the $n$ embedding tokens (n = N+1, including class token) as $Q = [\mathbf{q_1}, \mathbf{q_2}, \cdots, \mathbf{q_n}]^\top$, $K = [\mathbf{k_1}, \mathbf{k_2}, \cdots, \mathbf{k_n}]^\top$, $V = [\mathbf{v_1}, \mathbf{v_2}, \cdots, \mathbf{v_n}]^\top$. The original correlation matrix between $Q$ and $K$ is denoted by $QK^\top = [\alpha_{i,j}]_{n \times n}$. In our proposed DCT method, we introduce three domain conditioning vectors $C_q \in \mathbb{R}^{1 \times d}$, $C_k \in \mathbb{R}^{1 \times d}$, and $C_v \in \mathbb{R}^{1 \times d}$ and append them to the the query, key, and value matrices, respectively, and obtain the following augmented query $\bar{Q} \in \mathbb{R}^{(n+1) \times d}$, key $\bar{K} \in \mathbb{R}^{(n+1) \times d}$, and value $\bar{V} \in \mathbb{R}^{(n+1) \times d}$:

$$\bar{Q} = \begin{bmatrix} Q \\ C_q \end{bmatrix}, \quad \bar{K} = \begin{bmatrix} K \\ C_k \end{bmatrix}, \quad \bar{V} = \begin{bmatrix} V \\ C_v \end{bmatrix}. \quad (3)$$

The correlation matrix between $\bar{Q}$ and $\bar{K}$ with domain conditioners is:

$$\bar{Q}\bar{K}^\top = \begin{bmatrix} \alpha_{1,1} & \alpha_{1,2} & \cdots & \alpha_{1,n} & \mathbf{q_1}C_k^\top \\ \alpha_{2,1} & \alpha_{2,2} & \cdots & \vdots & \mathbf{q_2}C_k^\top \\ \vdots & \vdots & \ddots & \vdots & \vdots \\ \alpha_{n,1} & \alpha_{n,2} & \cdots & \alpha_{n,n} & \mathbf{q_n}C_k^\top \\ C_q\mathbf{k_1}^\top & C_q\mathbf{k_2}^\top & \cdots & C_q\mathbf{k_n}^\top & C_qC_k^\top \end{bmatrix} \quad (4)$$

Now, the new self-attention weight matrix $W = [w_{i,j}]_{(n+1) \times (n+1)}$ is given by:

$$w_{i,j} = \text{softmax}\left(\bar{Q}\bar{K}^\top\right)$$

$$= \begin{cases} \frac{\exp(\alpha_{i,j})}{\sum_{j=1}^n \exp(\alpha_{i,j}) + \exp(\mathbf{q_i} C_k^\top)}, & i \neq n+1, j \neq n+1; \\[2ex] \frac{e^{\mathbf{q_i} C_k^\top}}{\sum_{j=1}^n \exp(\alpha_{i,j}) + \exp(\mathbf{q_i} C_k^\top)}, & i \neq n+1, j = n+1; \\[2ex] \frac{\exp(C_q \mathbf{k_j}^\top)}{\sum_{j=1}^n \exp(C_q \mathbf{k_j}^\top) + \exp(C_q C_k^\top)}, & i = n+1, j \neq n+1; \\[2ex] \frac{\exp(C_q C_k^\top)}{\sum_{j=1}^n \exp(C_q \mathbf{k_j}^\top) + \exp(C_q C_k^\top)}, & i = n+1, j = n+1. \end{cases} \quad (5)$$

The conditioned self-attention output is:

$$\text{Attention}(\bar{Q}, \bar{K}, \bar{V}) = \text{softmax}\left(\frac{\bar{Q}\bar{K}^\top}{\sqrt{d}}\right)\bar{V}. \quad (6)$$

From (5) and (6), we can see that the original self-attention weights have been modified by the domain conditioner. The introduction of domain conditioners adds an additional context to the attention mechanism. The softmax score of $\alpha_{i,j}$ is re-calibrated to consider both input data and the contextual information of the target domain. In our proposed DCT method for fully test time adaptation, these domain conditioners $[C_q, C_k, C_v]$ are generated by a network that is learned online during the test-time adaptation process, which will be explained in the following section.

## 3.3 Domain Conditioner Generator Networks

At each transformer network layer, we introduce a dedicated light-weight network $\Phi_l$, called *domain conditioner generator* to generate the three domain conditioners $[C_q^l, C_k^l, C_v^l]$ from the class token $C^l$:

$$[C_q^l, C_k^l, C_v^l] = \Phi^l(C^l). \quad (7)$$

This domain conditioner generator network is learned during the test-time adaptation process. Specifically, in the current mini-batch $\mathbf{B}_t$, when training the network $\Phi_l$, we aim to minimize the loss function $\mathcal{L}(\theta_t; x)$ with respect to the learnable weights $\theta_t$ of network $\Phi_l$. The loss function $\mathcal{L}(\theta_t; x)$ in test-time adaptation is commonly defined by the entropy of the given batch. In practice, we find such minimization will cause model collapse. To address this issue, we use the reliable entropy minimization along with the sharpness-aware minimization [7, 38]. The reliable entropy minimization filters out testing samples with relatively large entropy to reduce the impact of noisy samples on the model's fine-tuning and makes it more robust to incomplete or noisy data. The sharpness-aware entropy minimization encourages the model weights to converge to a flat minimum, indicating that the model is robust to small perturbations in the weights. The overall optimization loss is defined as:

$$\mathcal{L}(\theta_t; x) = \mathbb{I}[\mathbb{E}(\theta_t; x) < E_0] \cdot \mathbb{E}(\theta_t; \mathbf{B}_t), \quad (8)$$

where $\mathbb{I}[\mathbb{E}(\theta_t; x) < E_0]$ is the mask to filter out test samples when entropy is larger than the threshold $E_0$, and $\mathbb{E}$ is the entropy function.

Figure 4 shows the t-SNE plot visualization of the domain conditioners $[C_q^l, C_k^l, C_v^l]$ in layer 1 for samples from different target domains, with different domain corruptions, plotted with three different colors. We can see that, samples from the same domain, although from totally different classes, aggregate together. This suggests that the domain conditioners $[C_q^l, C_k^l, C_v^l]$ generated by the learned network $\Phi^l$ are able to capture the domain characteristics.

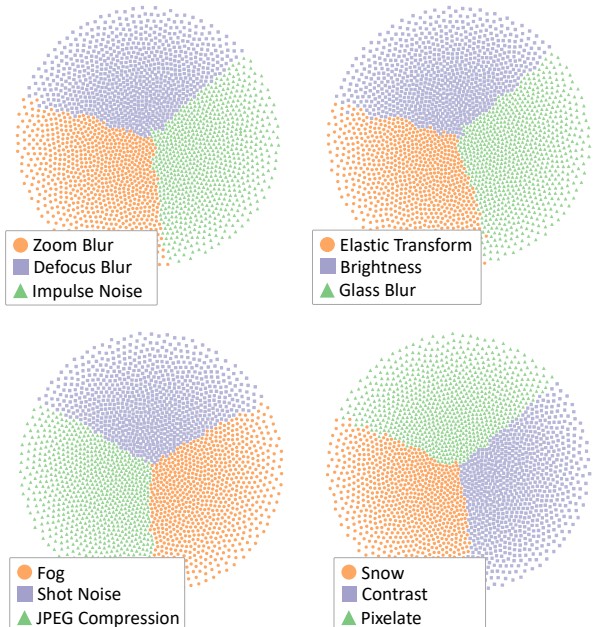

**Figure 4: Visualization of the domain conditioners for different domains in ImageNet-C from the first vision transformer layers.**

We observe that the domain conditioners play a crucial role in capturing and gradually removing the domain perturbation from the image features throughout the transformer network layers. Figure 5 shows the class tokens $C^l$ at different layers of our domain-conditioned transformer. Specifically, the first 5 plots show the class token at layers 1, 3, 6, 11, and 12 of the domain-conditioned transformer. Each plot shows the samples from 5 different target domains (Gaussian Noise, Frost, Defocus Blur, Contrast, and Fog) with each domain being plotted with a different color. We can see that, with the proposed domain conditioning learning and adaptation, the domain information is being gradually removed from the class tokens. In the 5-th plot for layer 12, we can hardly see any domain difference among these samples. For comparison, in the 6-th plot, we also show the class token of layer 12 from the source model without using the proposed DCT method. We can see that the domain information is clearly seen in the final layer of the transformer model. This will significantly degrade the performance of the network in the target domain.

## 4 EXPERIMENTS

In this section, we conduct experiments on multiple online test-time adaptation settings and multiple dataset benchmarks to evaluate the performance of our proposed DCT method.

## 4.1 Benchmark Datasets and Baselines.

In our experiments, we select the widely used **ImageNet-C** benchmark dataset [17], consisting of $50,000$ instances distributed across $1,000$ classes. Additionally, we test in **ImageNet-R** [16], a dataset containing 30,000 images presenting diverse artistic renditions of 200 classes from the ImageNet dataset. The results also include

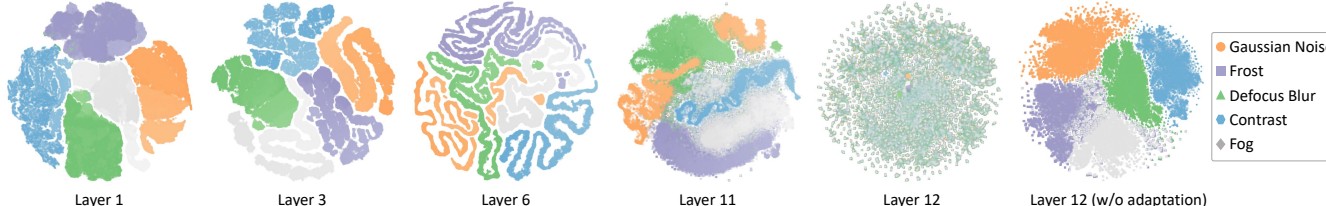

**Figure 5: Visualization of output class tokens from different vision transformer layers. The first 5 plots show the features from various layers of our DCT, and the last plot shows the features of the source model for comparison.**

**Table 1: Classification Accuracy (%) for each corruption in ImageNet-C under Normal at the highest severity (Level 5). The best result is shown in bold.**

| Method | gaus | shot | impul | defcs | gls | mtn | zm | snw | frst | fg | brt | cnt | els | px | jpg | Avg. |
|---|---|---|---|---|---|---|---|---|---|---|---|---|---|---|---|---|
| Source | 46.9 | 47.6 | 46.9 | 42.7 | 34.2 | 50.5 | 44.7 | 56.9 | 52.6 | 56.5 | 76.1 | 31.8 | 46.7 | 65.5 | 66.0 | 51.0 |
| T3A [20] | 16.6 | 11.8 | 16.4 | 29.9 | 24.3 | 34.5 | 28.5 | 15.9 | 27.0 | 49.1 | 56.1 | 44.8 | 33.3 | 45.1 | 49.4 | 32.2 |
| CoTTA [53] | 40.3 | 31.8 | 39.6 | 35.5 | 33.1 | 46.9 | 37.3 | 2.9 | 46.4 | 59.1 | 71.7 | 55.5 | 46.4 | 59.4 | 59.0 | 44.4 |
| DDA [11] | 52.5 | 54.0 | 52.1 | 33.8 | 40.6 | 33.3 | 30.2 | 29.7 | 35.0 | 5.0 | 48.6 | 2.7 | 50.0 | 60.0 | 58.8 | 39.1 |
| MEMO [58] | 58.1 | 59.1 | 58.5 | 51.6 | 41.2 | 57.1 | 52.4 | 64.1 | 59.0 | 62.7 | **80.3** | 44.6 | 52.8 | 72.2 | 72.1 | 59.1 |
| AdaContrast [3] | 54.4 | 55.8 | 55.8 | 52.5 | 42.2 | 58.7 | 54.3 | 64.6 | 60.1 | 66.4 | 76.8 | 53.7 | 61.7 | 71.9 | 69.6 | 59.9 |
| CFA [23] | 56.9 | 58.0 | 58.1 | 54.4 | 48.9 | 59.9 | 56.6 | 66.4 | 64.1 | 67.7 | 79.0 | 58.8 | 64.3 | 71.7 | 70.2 | 62.4 |
| TENT [51] | 57.6 | 58.9 | 58.9 | 57.6 | 54.3 | 61.0 | 57.5 | 65.7 | 54.1 | 69.1 | 78.7 | 62.4 | 62.5 | 72.5 | 70.6 | 62.8 |
| DePT-G [12] | 53.7 | 55.7 | 55.8 | 58.2 | 56.0 | 61.8 | 57.1 | 69.2 | 66.6 | 72.2 | 76.3 | 63.2 | 67.9 | 71.8 | 68.2 | 63.6 |
| SAR [38] | 58.0 | 59.2 | 59.0 | 58.0 | 54.7 | 61.2 | 57.9 | 66.1 | 64.4 | 68.6 | 78.7 | 62.4 | 62.9 | 72.5 | 70.5 | 63.6 |
| **Ours** | **58.8** | **60.2** | **60.1** | **58.7** | **58.9** | **63.2** | **62.9** | **69.4** | **68.1** | **73.2** | 79.6 | **65.1** | **69.0** | **74.4** | **72.3** | **66.3** |
| | ±0.2 | ±0.1 | ±0.1 | ±0.1 | ±0.3 | ±0.1 | ±0.2 | ±0.2 | ±0.3 | ±0.0 | ±0.1 | ±0.2 | ±0.1 | ±0.1 | ±0.2 | ±0.0 |

**VisDA-2021** [2], a dataset designed to assess models' ability to adapt to novel test distributions and effectively handle distributional shifts. Additionally, we utilize the **Office-Home** [50] dataset, which has a total of 15, 500 images spanning 65 object categories across four distinct domains. We compare our proposed DCT method against the following fully online test-time adaptation methods: no adaptation which is the source model, T3A, CoTTA, DDA, MEMO, TENT, AdaContrast, CFA, DePT-G, and SAR.

### 4.2 Implementation Details

Following the official implementations of SAR, we use the ViT-B/16 backbone for all experiments unless explicitly stated otherwise. The pre-trained model weights are obtained from the *timm* repository [54]. Specifically, for the Office-Home dataset, we fine-tune the ViT-B/16 model by replacing the original classifier head with a new classifier head. To ensure fair performance comparisons, all methods within each experimental condition share identical architecture and pre-trained model parameters. We employ the SGD optimizer with Sharpness Aware Minimization [8]. The batch size is set to 64 for all experiments, except for the condition *Batch size = 1*. Our reported experimental results are the mean and standard deviation values obtained from three runs, each with random seeds chosen from the set {2021, 2022, 2023}. It should be noted that we use the matched normalization setting for the pre-trained *timm* model (mean = [0.5, 0.5, 0.5], std = [0.5, 0.5, 0.5]), which is different from

the code of the original SAR paper [38]. All models are tested on a single NVIDIA RTX3090 GPU. The source code will be released.

### 4.3 Performance Results

We evaluate the performance of our DCT method under three different test conditions on the ImageNet-C dataset. We report the reproduced top-1 accuracy using the official codes for all methods under comparison. Specifically, (1) we first evaluate our approach under the **Normal** i.i.d assumption and compared it with other TTA methods. The results of this experiment are shown in Table 1. Our method outperforms other baseline methods for almost all 15 corruptions. On average, our method outperforms the second-best method by 2.7%. (2) Then, we evaluate our approach under the **Imbalanced label shifts** test condition with the same imbalanced sample sequence. The results of this experiment are shown in Table 2. We can see that our method improves the average classification accuracy of all 15 corruption types by 1.1%. (3) We evaluate our approach under the challenging **Batch size = 1** test condition, which is known to be particularly difficult for TTA methods. As shown in Table 3, our method improves the average classification accuracy of all 15 corruption types by 1.6%, demonstrating its superior robustness and adaptability with small batch sizes.

We also conduct experiments on the **ImageNet-R** and **VisDA-2021** datasets to verify the effectiveness of our method. For ImageNet-R, we use the same pre-trained ViT-B/16 backbone and set the output size to 200 following the procedure in [16]. From Table 5 and

**Table 2: Classification Accuracy (%) for each corruption in ImageNet-C under Imbalanced label shifts at the highest severity.**

| Method | gaus | shot | impul | defcs | gls | mtn | zm | snw | frst | fg | brt | cnt | els | px | jpg | Avg. |
|---|---|---|---|---|---|---|---|---|---|---|---|---|---|---|---|---|
| Source | 46.9 | 47.7 | 47.0 | 42.8 | 34.2 | 50.7 | 44.8 | 56.9 | 52.6 | 56.5 | 76.1 | 31.9 | 46.7 | 65.5 | 66.1 | 51.1 |
| DDA [11] | 52.6 | 54.0 | 52.2 | 33.7 | 40.8 | 33.6 | 30.2 | 29.8 | 35.0 | 5.0 | 48.8 | 2.7 | 50.2 | 60.2 | 58.9 | 39.2 |
| MEMO [58] | 58.1 | 59.1 | 58.5 | 51.6 | 41.2 | 57.1 | 52.4 | 64.1 | 59.0 | 62.7 | **80.3** | 44.6 | 52.8 | 72.2 | 72.1 | 59.1 |
| TENT [51] | 58.5 | 59.9 | 59.9 | 58.6 | 57.2 | 62.5 | 59.3 | 67.0 | 28.9 | 71.0 | 79.3 | 62.9 | 65.5 | 73.8 | 71.9 | 62.4 |
| SAR [38] | **59.0** | 60.2 | 60.1 | **59.0** | 57.6 | 62.7 | 59.7 | 67.5 | 66.2 | 70.6 | 79.4 | 63.1 | 66.3 | 73.7 | 71.9 | 65.1 |
| **Ours** | 58.8 | **60.5** | **60.2** | 58.9 | **58.6** | **63.6** | **62.6** | **69.1** | **68.3** | **72.8** | 79.5 | **63.9** | **69.1** | **74.3** | **72.5** | **66.2** |
|  | ±0.1 | ±0.1 | ±0.2 | ±0.1 | ±0.3 | ±0.1 | ±0.5 | ±0.2 | ±0.3 | ±0.2 | ±0.1 | ±0.8 | ±0.4 | ±0.4 | ±0.1 | ±0.1 |

**Table 3: Classification Accuracy (%) for each corruption in ImageNet-C under Batch size = 1 at the highest severity.**

| Method | gaus | shot | impul | defcs | gls | mtn | zm | snw | frst | fg | brt | cnt | els | px | jpg | Avg. |
|---|---|---|---|---|---|---|---|---|---|---|---|---|---|---|---|---|
| Source | 46.9 | 47.6 | 46.9 | 42.7 | 34.2 | 50.5 | 44.7 | 56.9 | 52.6 | 56.5 | 76.1 | 31.8 | 46.7 | 65.5 | 66.0 | 51.0 |
| DDA [11] | 52.5 | 54.0 | 52.1 | 33.8 | 40.6 | 33.3 | 30.2 | 29.7 | 35.0 | 5.0 | 48.6 | 2.7 | 50.0 | 60.0 | 58.8 | 39.1 |
| MEMO [58] | 58.1 | 59.1 | 58.5 | 51.6 | 41.2 | 57.1 | 52.4 | 64.1 | 59.0 | 62.7 | **80.3** | 44.6 | 52.8 | 72.2 | 72.1 | 59.1 |
| TENT [51] | 58.6 | 60.1 | 60.0 | 59.0 | 57.4 | 62.7 | 59.7 | 67.3 | 45.5 | 71.4 | 79.2 | 63.9 | 66.1 | 73.9 | 71.9 | 63.8 |
| SAR [38] | 59.1 | 60.2 | 60.1 | 58.5 | 55.9 | 62.4 | 59.2 | 67.5 | 66.0 | 70.2 | 78.8 | 62.7 | 65.6 | 73.9 | 71.9 | 64.8 |
| **Ours** | **59.5** | **61.0** | **60.7** | **59.2** | **59.1** | **63.8** | **62.0** | **69.6** | **68.5** | **73.5** | 78.8 | **64.7** | **68.8** | **74.2** | **72.4** | **66.4** |
|  | ±0.1 | ±0.1 | ±0.1 | ±0.1 | ±0.1 | ±0.1 | ±0.2 | ±0.4 | ±0.3 | ±0.3 | ±0.2 | ±0.4 | ±0.1 | ±0.5 | ±0.3 | ±0.0 |

**Table 4: Classification Accuracy (%) for test-time adaptation in Office-Home dataset.**

| Method | A→C | A→P | A→R | C→A | C→P | C→R | P→A | P→C | P→R | R→A | R→C | R→P | Avg. |
|---|---|---|---|---|---|---|---|---|---|---|---|---|---|
| Source | 63.4 | 81.9 | 86.3 | 76.2 | 80.6 | 83.8 | 75.0 | 57.9 | 87.2 | 78.7 | 61.0 | 88.0 | 76.7 |
| TENT [51] | 69.1 | 81.8 | 86.5 | 76.5 | 81.9 | 83.2 | **76.8** | 65.0 | 86.7 | **81.1** | 69.7 | **88.2** | 78.9 |
| SAR [38] | 67.3 | 80.7 | 85.6 | 77.5 | 79.8 | 84.1 | 74.7 | 60.3 | 87.6 | 78.9 | 63.1 | 87.7 | 77.3 |
| **Ours** | **69.2** | **82.6** | **87.2** | **78.4** | **83.6** | **85.2** | **76.8** | 65.3 | **87.9** | 80.2 | 67.0 | 88.1 | **79.3** |
|  | ±0.1 | ±0.1 | ±0.1 | ±0.1 | ±0.1 | ±0.7 | ±0.0 | ±0.1 | ±0.0 | ±0.1 | ±0.1 | ±0.2 | ±0.1 |

**Table 5: Classification Accuracy (%) in ImageNet-R under Normal and Batch size=1 settings.**

| Method | Normal | Batch size = 1 |
|---|---|---|
| Source | 57.2 | 57.2 |
| TENT [51] | 61.3 | 61.5 |
| SAR [38] | 62.0 | 61.8 |
| **Ours** | **64.5** | **65.0** |
|  | ±0.2 | ±0.4 |

**Table 6: Classification Accuracy (%) in VisDA-2021 under Normal and Batch size=1 settings.**

| Method | Normal | Batch size = 1 |
|---|---|---|
| Source | 57.7 | 57.7 |
| TENT [51] | 60.1 | 60.1 |
| SAR [38] | 60.1 | 60.9 |
| **Ours** | **62.2** | **62.7** |
|  | ±0.2 | ±0.5 |

Table 6, we can see that the overall results are consistent with those on ImageNet-C. Our approach outperforms the previous state-of-the-art methods in both experimental settings, namely Normal and Batch size = 1. It demonstrates that the proposed DCT method is effective in different domains. We extend our experimentation to **Office-Home**. The results are presented in Table 4. The proposed DCT method outperforms the SAR method by 2.0%. This further

underscores the efficacy of our proposed DCT method across a diverse range of datasets.

Overall, our experimental results demonstrate the effectiveness and robustness of our proposed DCT approach in handling complex test conditions and outperforming state-of-the-art TTA methods across multiple evaluation metrics. More experimental results are provided in the Supplementary Materials.

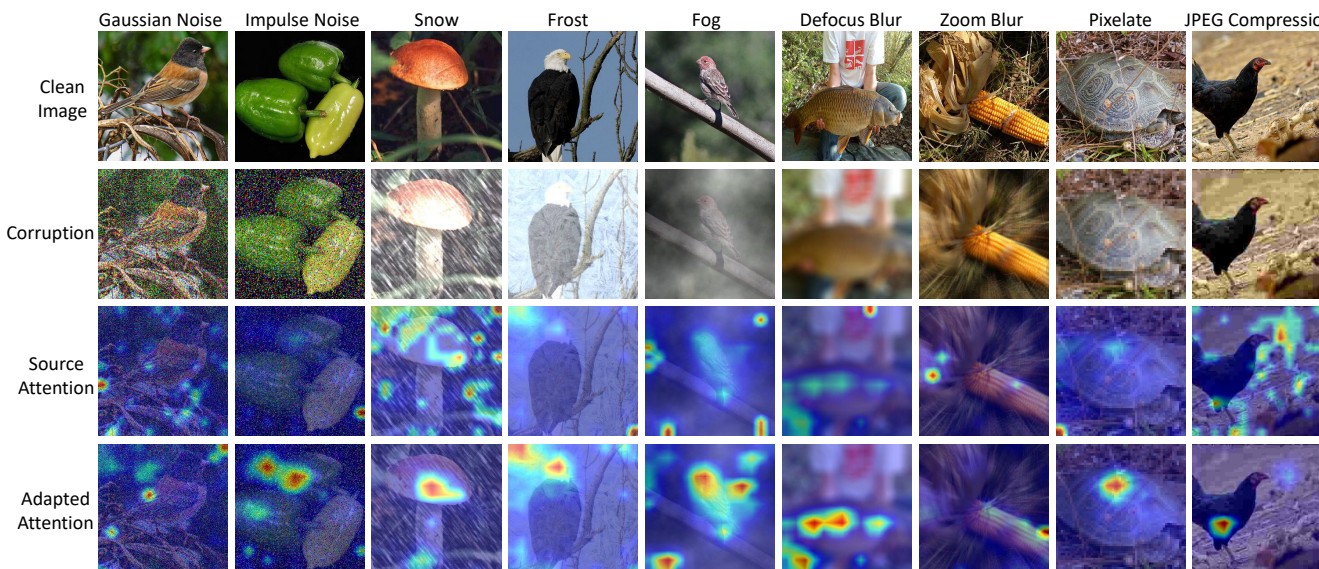

**Figure 6: Representative examples of Attention Rollout [1] for the source model (3-rd row) and our adapted model (last row).**

## 4.4 Visualization and Discussion

**Table 7: Ablation study under Normal at the highest severity. DC-generator represents the domain conditioner generator.**

| Methods | Avg. |
|---|---|
| Baseline Method | 63.6 |
| + Domain-conditioner w/o DC-generator | 63.9 |
| **Our DCT Method** | **66.3** |

To explore the explainability of the domain-conditioned transformer, we visualize the attention map by Attention Rollout [1] following ViT [6]. As shown in Figure 6, given the corruption image, we can see that the adapted transformer attention focuses more on the object than the source model. This demonstrates that our Domain-Conditioned Transformer (DCT) method significantly enhances the attention in the target domain.

Additionally, we performed an ablation study on the domain-conditioner generator shown in Table 7. When solely integrating learnable domain-conditioners into query, key, and value without the domain-conditioner generator's conditional generation based on the class token, the average accuracy improved by 0.3%. In contrast, when adapting the domain-conditioner generator to generate domain conditioners conditioned by the class token, we observed a substantial improvement of 2.7%. It demonstrates the significant contribution of the class token conditioned domain-conditioner generator in enhancing the model's performance.

We conduct parameter sensitivity analysis on the learning rates for the domain-conditioner generator with the Normal setting in ImageNet-C with ViT-B/16. As shown in Figure 7, we can see that the performance is best when the learning rate of the domain-conditioner generator is set to 0.01.

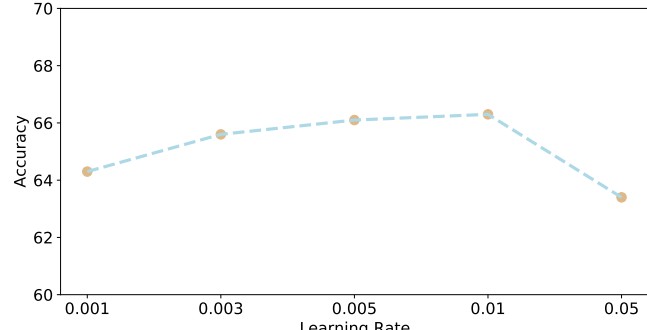

**Figure 7: Sensitivity analyses for the learning rates of the domain-conditioner generator.**

## 5 CONCLUSION

Fully test-time adaptation is a challenging problem in computer vision, particularly in the presence of complex corruptions and shifts in the test data distribution. In this work, we have tackled the critical challenge of adapting transformer-based models to new domains, focusing on the significant deviation in self-attention profiles encountered in the target domain compared to the source domain. Specifically, we have introduced three domain-conditioning vectors, called domain conditioners into the self-attention module. By integrating these domain conditioners into the query, key, and value components of the self-attention module, we have effectively mitigated the impact of domain shift observed during inference. The dynamic generation of these domain conditioners at each transformer network layer, derived from the class token, allowed for a gradual removal of domain shift effects, thereby enabling the recovery of the original self-attention profile in the target domain. Our experimental results demonstrated that our proposed DCT method is able to significantly improve fully test-time adaptation performance.

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
