# OpenReview forum: "Domain-Conditioned Transformer for Fully Test-time Adaptation"
_acmmm.org/ACMMM/2024/Conference — MM2024 Poster_

### Official Review · Reviewer_QvLt · 2024-05-09

**Rating:** 3
**Confidence:** 3

**Summary:**

This paper proposes a Domain-Conditioned Transformer for fully test-time adaptation, introducing domain-conditioning vectors into the self-attention mechanism. During test time, a network is learned to generate these domain conditioners. Experimental results and analysis demonstrate the effectiveness of the proposed approach.

**Strengths:**

1. The authors conduct extensive experiments on various online test-time adaptation settings across multiple dataset benchmarks, demonstrating that the proposed model outperforms previous SOTA method SAR.
2. The visualizations of output class tokens in Figure 1 and Figure 5 exhibit that the proposed method effectively reduces the inter-domain distance while increasing the inter-class distance,  thereby mitigating the influence of domain gap. Figure 6 also shows improvements in attention within the target domain.

**Limitations:**

1. The proposed method need further explanation. The domain conditioners, which are actually prompts generated from the class tokens, require a clearer illustration regarding how these vectors leverage information from class tokens to mitigate the impact of domain shift. It would be also beneficial for the authors to further discuss the relationship with prompt learning.
2. In Section 3.1 and Figure 2, the authors investigate the self-attention profile. There is confusion regarding why domain shift results in larger attention distances and how increased attention distance affects performance.
3. Figure 4 shows the t-SNE visualization of the domain conditioners, but I am wondering that whether different domains can be separated so distinctly. Is there any post-processing applied? I recommend the authors provide a detailed illustration of the visualization procedure.

**Suitability:**

2

---

### Official Review · Reviewer_w6ch · 2024-05-18

**Rating:** 3
**Confidence:** 3

**Summary:**

In this paper, the authers explore fully online test-time adaptation of vision transformer models.
They introduce a new design of the self-attention module in the transformer, which can capture the domain-specific characteristics of test samples in the target domain and is able to clean up the domain perturbations in the test samples.
To be specific, they learn a lightweight neural network called domain-conditioner generator to generate the domain conditioners from the class token at each layer, and incorporate the three domain conditioners into the query, key, and value components of the self-attention module, respectively.
Experimental results demonstrate effictiveness.

**Strengths:**

1) The method  improves the online fully test-time domain adaptation performance and outperforms existing state-of-the-art methods.
2) The running time cost of the proposed approach is promising.
3) The paper is good writing and has elegant figures and tables.

**Limitations:**

1) what is the complexity and structure of the domain conditioner generator? It is crucial to know model size of DC-generator because the component plays a major role in performance improvement according to Table 7.
2) There are some typos, e.g. excess "the" in the line 448.
3) The description of figures is not clear. For instance, the authers need to state that "Gaussian Noise", "Defocus Blur" and "Snow" indicate domains in Figure 1.

**Suitability:**

3

---

### Official Review · Reviewer_KAnN · 2024-05-23

**Rating:** 3
**Confidence:** 2

**Summary:**

This paper focuses on fully test-time adaptation (TTA). The proposed domain-conditioned transformer construct domain-conditioners to capture domain-specific perturbation information and remove these perturbations layer by layer. In addition, domain-conditioned transformer learns a domain conditioner generation network to generates the abovementioned domain conditioners, which contains semantic and domain information. Experimental evaluation shows the effectiveness of the proposed method.

**Strengths:**

*	The idea seems interesting for dealing with TTA.
*	The paper did thorough experiments across multiple benchmarks with performance gains.

**Limitations:**

*	The motivation behind the adopted method, particularly regarding domain-conditioned self-attention, is not sufficiently clarified. It is essential to provide a more comprehensive explanation of the rationale behind this choice to strengthen the foundation of the proposed approach.
*	The implementation details of the domain-conditioner generator network are missing. The authors are suggested to provide more technical details about how to implement the domain-conditioner generator network.
*	In Tab. 7, what dataset was used to illustrate the ablation study results? Additionally, the backbone utilized in the ablation study was not provided.
*	The experiments on SAR [1] are all based on the ImageNet-C dataset. If the experimental results of SAR are reproduced by the public code, please given the relevant elucidations.

[1] Niu et al. Towards stable test-time adaptation in dynamic wild world. In ICLR, 2023.

**Suitability:**

2

---

### Meta-Review · Area_Chair_Ga2Q · 2024-06-29

**Recommendation:** Accept (Poster)
**Confidence:** 4

**Metareview:**

The paper initially got three borderline rejections. The authors have provided a rebuttal. After checking the rebuttal, two of the reviewers are satisfied with the rebuttal and changed their ratings to borderline accepts. However, the other reviewer kept the original borderline rejection, as he/she thinks the description of the domain conditioner generator network is not clear enough. The AC has carefully checked the rebuttal about the domain conditioner generator network and found that the authors have already clearly described it. Hence, the AC thinks all the concerns of the reviewers have been solved during rebuttal and this is a borderline accept paper. In the camera ready, the AC strongly encourages the authors to include all discussions in the rebuttal, provide more details of important parts, and release the source code.